

# Regional dynamical and statistical downscaling temperature, humidity and windspeed for the Beijing region under stratospheric aerosol injection geoengineering

Jun Wang[1], John C. Moore[1,2*], Liyun Zhao[1*], Chao Yue[1], Zhenhua Di[3]

[1]College of Global Change and Earth Systems Science, Beijing Normal University, Beijing, China
[2]Arctic Center, University of Lapland, Rovaniemi, Finland
[3]State Key Laboratory of Earth Surface Processes and Resource Ecology, Faculty of Geographical Science, Beijing Normal University, Beijing 100875, China

*Correspondence to:* zhaoliyun@bnu.edu.cn, john.moore.bnu@gmail.com

**Abstract.** We use four Earth System Model (ESM) to simulate climate under the modest greenhouse emissions RCP4.5, the "business-as-usual" RCP8.5 and the stratospheric aerosol injection G4 geoengineering scenarios. These drive a 10 km resolution dynamically downscaled model (WRF), and a statistically bias corrected (ISIMIP) and downscaled simulation in a 450×330 km domain containing the Beijing province, ranging from 2000 m elevations to sea level. The 1980s simulations of surface temperatures, humidities and windspeeds using statistical bias correction makes for a better estimate of mean climate determined by ERA5 reanalysis data than does the WRF simulation. However correcting WRF output with Quantile-Delta Mapping bias correction removes the offsets in mean state and results in WRF better reproducing observations over 2007-2017 than ISIMIP bias correction. WRF simulations show consistently 0.5°C higher mean annual temperatures than from ISIMIP due both to the better resolved city centers and also to warmer winter temperatures. In the 2060s WRF produces consistently larger spatial ranges of surface temperatures, humidities and windspeeds than ISIMIP downscaling across the three future scenarios. WRF and ISIMIP methods produce very similar spatial patterns of temperature with G4 are always cooler than RCP4.5 and RCP8.5, by a slightly larger amount with ISIMIP than WRF. Humidity scenario differences vary greatly between ESM and hence ISIMIP downscaling, while for WRF the results are far more consistent across ESM and show only small changes between scenarios. Mean windspeeds show similarly small changes over the domain, although G4 is significantly windier under WRF than either RCP scenario.



## 1 Introduction

The global-mean surface air temperature has increased by 0.9°C-1.2°C relative to 1850-1900 (Eyring et al., 2021), with a rapid rise during the 2010s. Extreme climate events are becoming more frequent
(Pachauri et al., 2014), impacting human health and mortality rates (Pielke et al., 2013). ESM despite global in extent, cannot simulate phenomena smaller than their spatial resolution (typically 1-2°) with the same fidelity as higher resolution models with much smaller domains. Higher resolution models include regional climate models and the Weather Research and Forecasting model (WRF) which are generally driven by ESMs at their lateral boundaries. WRF has been widely used as a dynamical
downscaling method for future climate projection at small and regional scales (Bao et al., 2015; Brewer and Mass et al, 2016; Kong et al., 2019). Kong et al. (2019) found that WRF was satisfactory in reproducing spatiotemporal distribution and trends of extreme climate indices for China.

Geoengineering via increasing planetary albedo as a method of avoiding the worst excesses of climate
heating has been actively discussed in climate research for well over a decade (Shepherd, 2009). The most widely studied albedo modification type (e.g. Lenton et al., 2009; Robock et al., 2009) is via stratospheric aerosol injection (SAI). To standardize and aid the evaluation of SAI in ESM simulations, Kravitz et al. (2011) proposed the Geoengineering Model Intercomparison Project (GeoMIP), with Phase 1 including two different SAI scenarios using sulfates as the aerosol, and with greenhouse gas
emissions from the Representative Concentration Pathway (RCP) 4.5 scenario. The impacts of SAI on temperature (Schmidt et al., 2012), precipitation (Tilmes et al., 2013), and the cryosphere (Moore et al., 2019) show that indeed the global mean temperatures are reduced, albeit with imperfections such as relative over-cooling of the tropics and under-cooling of the polar regions, and with relatively modest impacts on precipitation, especially compared with the less mitigated greenhouse gas scenarios. Several
studies have considered global-scale impacts on temperature and precipitation extremes under both SAI and other geoengineering types designed to enhance planetary albedo (Curry et al., 2014; Aswathy et al., 2015; Ji et al., 2018), and some studies have focused on regional impacts such as in Europe (Jones et al., 2018), East Asia (Kim et al., 2020), or the Maritime Continent (Kuswanto et al., 2021).



Statistical downscaling has often been used as an alternative to dynamical methods, avoiding the significant computing resources needed to run models such as WRF. Statistical downscaling is based on the relationships found historically between ESM output and observed climate variables and is very widely used in regional impact studies (Wilby et al., 2004). All models produce results with a bias from observations, and future simulations require either bias correction, or results are shown as climate anomalies relative to some control scenario. The ISIMIP (https://www.isimip.org/) consortium has produced methods (Hempel et al., 2013) widely used to correct the bias from CMIP5 (Climate Model Intercomparison Project phase 5) and GeoMIP outputs (McSweeney et al., 2016; Moore et al., 2019; Kuswanto et al., 2021). In our paper, we compare ISIMIP statistical downscaling methods and output from WRF dynamical downscaling and assess their performance for simulating the climate condition in the provinces around Beijing under SAI.

The greater Beijing region lies in complex terrain, surrounded by hills and mountains on three sides, with a flat plain to the southeast coast (Fig. 1). We explore the effect of geoengineering on surface temperature, wind and humidity in this domain. We nest a 10 km resolution domain inside a much larger 30 km resolution domain, driven at the boundaries with ESM output. We use the WRF model to dynamically downscale three time slices: 1979-1989, 2007-2017 and 2059-2069 driven by four ESM simulating the Historical, RCP 4.5, RCP8.5, and the GeoMIP G4 scenarios (Table 1). The 10 km resolution we use is not designed to study urban processes. Instead, we examine differences in downscaling at resolutions higher than, but comparable with, statistical downscaling methods that are likely to continue to be used in most geoengineering studies globally. To the best of our knowledge, this paper is the first to make dynamic downscaling of geoengineering scenarios.

**Table 1.** The simulations in our study.

| Periods | Scenarios | Goal of the simulation |
| --- | --- | --- |
| 1979-1989 | Historical | Compare ISIMIP statistical downscaling and bias-correction with WRF |
| 2007-2017 | RCP4.5 | Assess the performance of ISIMIP and WRF with bias-correction |
| 2059-2069 | RCP4.5, RCP8.5, G4 | Future downscaled climates by ISIMIP and WRF with bias-correction |



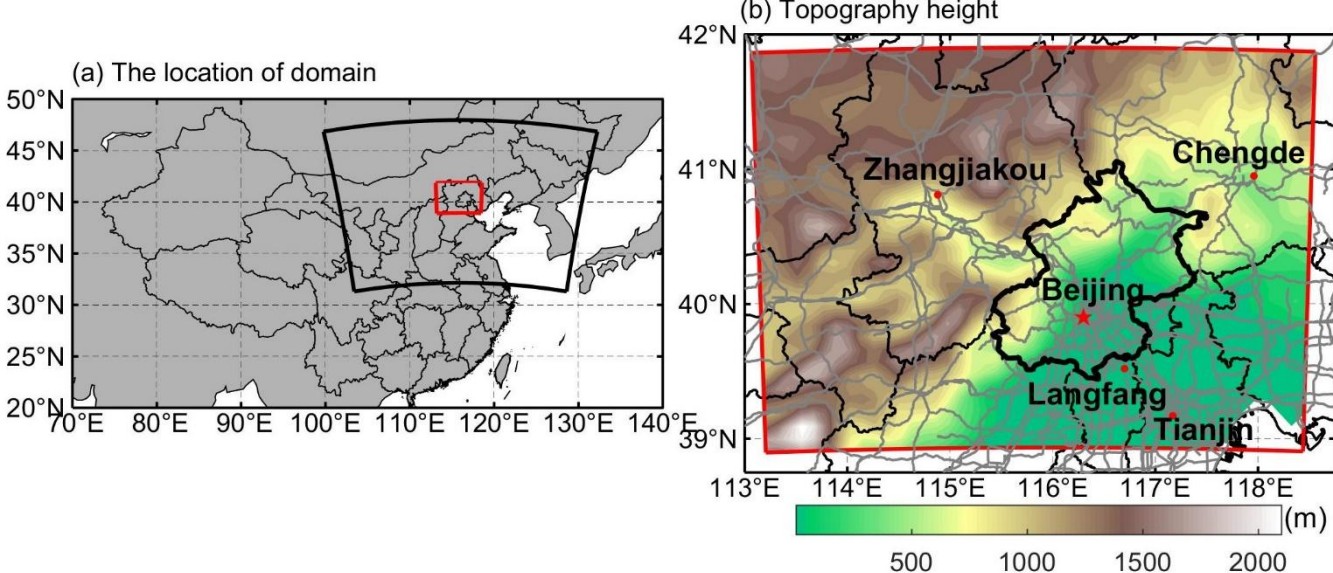

**Figure 1.** (a) Map of East Asia with Chinese provincial boundaries marked in black. The 10 km WRF domain (red box) is nested inside the 30 km resolution domain (large black sector), which is centered on 116°E, 40°N on a Lambert projection. (b) The topography and primary roads (gray curves) of the 10 km resolution domain from panel (a). The provincial boundaries are marked in black with the heavier line demarking the Beijing province. The major metropolitan centers of Beijing, Tianjin, Chengde and Langfang are marked in red.

We firstly show the differences between statistical downscaling with bias correction and dynamical downscaling without bias correction in the 1979-1989 period. This will show that statistical downscaled and bias corrected result, by design, has closer agreement with observations, despite its absence of physics, than the dynamically downscaled simulation. For the recent past period 2007-2017 we use the quantile delta mapping method to statistically correct the bias of WRF simulation and assess its performance. Finally, we show the projections in the future period 2059-2069 for the greater Beijing region where global temperature differences under different greenhouse gas and G4 scenarios are known to be large. The paper is organized as follows: Section 2 describes the WRF model setup and parameterization, statistical downscaling and bias correction methods. The results from the historical simulation and future projections on the surface temperature, humidity, and wind speed are all given in Section 3. Finally, a summary of the main findings and conclusion is given in Section 4.





## 2 Data and Method

### 2.1 ESMs and Scenarios (Data Description)


We focus on exploring the effect of SAI on surface meteorological conditions (temperature, humidity, and wind) over the domain using four different dynamically and statistically downscaled ESMs. In the simulations, we use three different scenarios: RCP4.5 and RCP8.5 and the GeoMIP G4 scenario. RCP4.5 is a scenario that never exceeds a radiative forcing of 4.5 W m$^{-2}$ (Thomson et al., 2011), while


RCP8.5 is an unmitigated emissions scenario leading to a radiative forcing of 8.5 W m$^{-2}$ at the end of the 21$^{st}$ century (Riahi et al., 2011). The GeoMIP experiment G4 specifies injection of sulfur dioxide into the equatorial lower stratosphere at a rate of 5 Mt per year from 2020 to 2069 (Kravitz et al., 2011). These scenarios span a useful range of climate scenarios: RCP4.5 is similar (Vandyck et al., 2016) to the expected trajectory of emissions under the 2015 Paris Climate Accord Nationally Determined


Contributions (NDCs); RCP8.5 represents a formerly business-as-usual scenario that still provides a large signal to noise ratio "worst case" scenario; G4 represents a similar radiative forcing as a quarter of the 1991 Mount Pinatubo volcanic eruption every year. If SAI were ever done then it would certainly use a much more sophisticated injection procedure than G4, perhaps designed to maintain hemispheric temperature balance and preserve pole-equator temperature gradients (Macmartin and Kravitz, 2016).


However, G4 levels of SAI are within the linear response of temperature reduction to material injected, and within a range of radiative forcing that might be plausible or reasonable to consider (Niemeier and Timmreck, 2015). GeoMIP has also developed new experiments for use with CMIP6 level ESMs (Kravitz et al., 2015).


ESM data required as input data for WRF includes meteorological fields, land surface and soil properties: specific humidity, air temperature, eastward wind, northward wind, near surface air pressure and the elevations of 30 pressure levels from 1000 hPa to 30 hPa, soil temperatures, humidities and water contents and sea level pressures. Only four ESMs can meet the data requirements (Table 2). We only use one single realization (r1i1p1 using the CMIP5 nomenclature) for each model since all


downscaling runs are extremely computationally expensive and some of the models have only a single realization available.





**Table 2.** ESMs used in this study.

| Model | Resolution (lon×lat) | Reference |
|---|---|---|
| BNU-ESM | 128×64 | Ji et al. (2014) |
| HadGEM2-ES | 192×144 | Collins et al. (2011) |
| MIROC-ESM | 128×64 | Watanabe et al. (2011) |
| MIROC-CHEM-ESM | 128×64 | Watanabe et al. (2011) |

We also use the 31 km resolution ERA5 6-hourly reanalysis data during 1979/01/01-1989/12/31 to

correct ESM climate fields at the domain boundaries as required by WRF (Hersbach et al., 2018). ERA5 reanalysis near surface meteorological elements (2 m temperature, 2 m humidity and 10 m wind speed) are significantly correlated with observations over the area (Meng et al., 2018). We use daily temperature, humidity and wind from ERA5 for the period 1980-1989 and 2008-2017 to statistically bias correct the ESMs variables and assess the performance of WRF downscaling.

**2.2 WRF**

The WRF model adopts a compressible non-hydrostatic equilibrium equation, and a variety of physical parameterization schemes and data assimilation which can realize high-resolution weather forecasts at various scales (Michalakes et al., 2001). Xu et al. (2012) used WRF to improve the bias in a global climate model simulations of extreme weather events. WRF needs 6 hourly input data, which does not

exist for most variables in the ESMs climate simulation output. So, we use the available monthly ESM data to estimate 6-hourly input data with a pseudo global warming downscaling method (PGW-DS, Kawase et al., 2009) using ERA5 data bilinearly interpolated to the same grid as the ESM output:

$$M_f = R_h - \overline{M_h} + \overline{M_f} \qquad\qquad (1)$$

where $M_f$ is the ESM-driven WRF model 6-hourly input data, $R_h$ is 6-hourly data from ERA5

reanalysis during the historical period, 1979-1989. For the period 1979-1989, $\overline{M_h}$ is the monthly data of ERA5, $\overline{M_f}$ is the monthly data of ESMs under historical scenario. For the period 2007-2017, $\overline{M_h}$ is the monthly data of ESMs during 1979-1989 under historical scenario, and $\overline{M_f}$ takes the monthly data of ESMs during 2007-2017 under RCP4.5. For the period 2059-2069, $\overline{M_h}$ is the monthly data of ESMs during 1979-1989 under historical scenario, and $\overline{M_f}$ takes the monthly data of ESMs during 2059-2069

under RCP4.5, RCP8.5 and G4 scenarios.



This study uses WRF version 3.9.1 with 2 nested domains, where the inner domain (longitude × latitude: 45×33) has a resolution of 10 km and the outer domain (80×58) has a resolution of 30 km (Fig. 1). The model has 30 vertical layers from the surface to 50 hPa. The integration timestep is 3 minutes. We set
the parameterizations following a study on the numerical simulation of urbanization on regional climate in China (Wang et al., 2012) as follows: the WRF Single Moment 6-class (WSM6, Hong et al. 2006), the new version of rapid radiative transfer model (RRTMG, Iacono et al., 2008) for both the long-wave and short-wave radiation, the MM5 similarity surface layer scheme (Paulson et al, 1970), the Yonsei University (YSU) planetary boundary layer (PBL) scheme (Noh et al., 2003), the Kain-Fritch scheme
for atmospheric convection (Kain, 2004), and the Noah land surface model (Chen and Dudhia, 2001). Putting the ESMs data as initial and boundary conditions into the WRF Preprocessing System (WPS; Fig. 2) is challenging. We downloaded all the required monthly data (See Table S1) from 4 ESMs in the 3 periods (1979-1989, 2007-2017, 2059-2069) and 6-hourly historical data from ERA5 (1979-1989). Then, we used the PGW method to get the 6-hourly input data during three periods (see Equation 1).
We then used Climate Data Operators (CDO) to convert the input data in NC format files into GRIB format files that WPS can recognize. In all three simulated periods, the initial year is considered as spin-up time and is not included in our analysis.

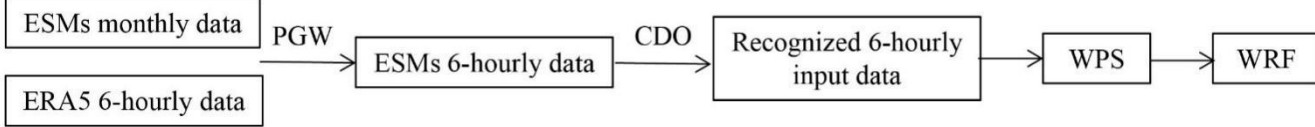

**Figure 2.** The WRF flow chart. PGW refers to the pseudo global warming downscaling method, and
CDO refers to Climate Data Operators used for generating the WRF recognizable "GRIB" format input data. WPS is the WRF Preprocessing System.

**2.3 ISIMIP statistical downscaling and bias correction**

This method corrects daily variability on the premise that the monthly trend of the modeled variable is
unchanged (Hempel et al., 2013) and includes two steps:





Step 1: Monthly bias corrected data is found by annual average difference between the model output and reanalysis data.

Step 2: Bias corrected daily data can be calculated by adding a linear regression residual:

$$M_d^*=(\overline{R_m}-\overline{M_m})+M_m+\overline{B}\times(M_d-M_m). \qquad (2)$$

$M_d^*$ is the bias corrected daily data. $\overline{R_m}$ and $\overline{M_m}$ are multi-year averaged values in one specific month from reanalysis data and model data, respectively. $M_m$ and $M_d$ are the monthly and daily model output in the specific month. The subscripts $m$ and $d$ represent monthly and daily, respectively. $\overline{B}$ is the linear regression coefficient of daily residual values between observed data and model data. Here, we use the ERA5 reanalysis data as observed data in our study. For convenience we use the term ISIMIP-ESM to

denote the output from the ESMs after applying the ISIMIP statistical downscaling and bias correction methodology.

## 2.4 Quantile Mapping (QM) and Quantile delta Mapping (QDM)

Quantile mapping has been widely used as a statistical bias-correction and downscaling method (Li and

Babovic, 2019; Kuswanto et al., 2021), and annual and monthly biases of all variables can be reduced to nearly zero (Wilcke et al., 2013). As a bias correction method, quantile mapping can reproduce the frequency of different types of extreme heat wave events well (Schoof et al., 2019). Here we use the empirical CDF to correct the biases:

$$M_d^*=F_R^{-1}(F_H(M_d)). \qquad (3)$$

$M_d^*$ is the daily data after bias correction, $F$ is the cumulative distribution function (CDF) and $F^{-1}$ is the inverse, subscript $R$ represents the ERA5 reanalysis data, subscript $H$ represents historical simulation results, and $M_d$ is daily model output in the historical simulations. This method keeps the model and observational data CDFs as consistent as possible.

QDM is similar to QM but is non-stationary. It considers the time variability between the historical simulation and future projection, hence it is preferable for our task here (Salvi et al., 2011).

$$M_d^*=F_R^{-1}\left(F_F(M_d)\right)+\left(M_d-F_H^{-1}\left(F_F(M_d)\right)\right). \qquad (4)$$



The $M_d^*$ and $M_d$ are bias corrected and raw daily model output in the future simulations. The subscript $R$ represents the ERA5 reanalysis data, the subscript $F$ and $H$ represents model outputs from future and historical simulations, respectively. To preserve the spatial information of the high resolution WRF model result, we do bias correction on daily value averaged in the whole inner domain (Fig. 1b) rather than separately for each grid point.

For the WRF simulations during 2007-2017 and 2059-2069, we use the QDM method to correct biases. Similar to ISIMIP-ESM, we use the terms WRF-ESM and WRF-QDM-ESM to represent results of WRF driven by the ESM and WRF driven by ESM after QDM bias correction, respectively.

## 3 Results

### 3.1 Historical simulation: WRF and ISIMIP downscaling comparison

We compare the simulations of mean temperature, relative humidity, and wind speed from raw ESM output downscaled to ERA5 resolution, ISIMIP-ESM and WRF-ESM in Beijing during the 1980s in Fig. 3. Fig. 3a shows the annually averaged 2 m ERA5 temperatures in Beijing, with a mean of 7 °C and highs in the southeast (12 °C) and lows in the northwest, which correlates with the topography (Fig. 1). Relative humidity (Fig. 3e) varies between 50%-55%, with the city center a little drier than the suburbs. Similarly, wind speed is low in the city center with highs in the higher north-western hills and south-eastern plain (Fig. 3i).





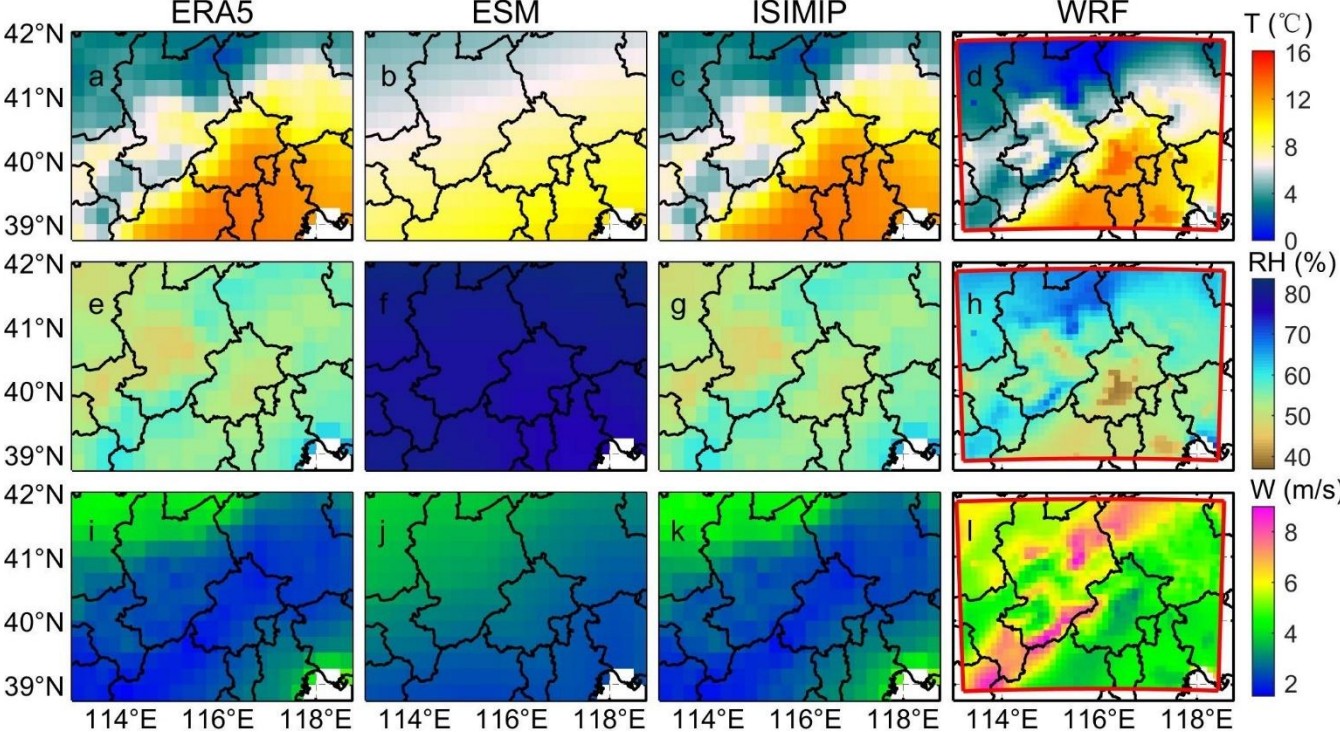

**Figure 3.** The region containing the high resolution WRF domain (red boundary in right column maps), with city boundaries marked in black. The spatial distribution of mean 2 m temperature (a-d), relative humidity (e-h) and 10 m wind speed (i-l) from ERA5, downscaled ESM ensemble mean before any bias correction, ISIMIP-ESM multi-ensemble mean and the WRF-ESM multi-ensemble mean in the high-resolution domain (Fig. 1) during 1980-1989. The results for the four ESM are shown separately in Figs. S1-S3 along with the bias corrected versions.

The temperatures from the raw ESM outputs in Beijing have less range than both ISIMIP-ESM and WRF-ESM results due to their coarser resolution, and obvious bias. The mean temperatures of MIROC-ESM and MIROC-ESM-CHEM over the domain are about 8°C while HadGEM2-ES and BNU-ESM are cooler, and all ESM have a large cold bias compared with ERA5 (Fig. S1). ISIMIP-ESM forces the model mean data to agree with ERA5 mean observations by design, and also downscales the ESM data to the observational resolution (Fig. 3c, 3g, 3k). The resulting ISIMIP-ESM means are indistinguishable by eye from the ERA5 mean in Fig. 3, though the ESM trends over time are preserved and the seasonality and other measures of variability are ESM-dependent and differ from ERA5.



WRF has a finer resolution than ERA5 and clearly shows higher temperatures in the center of Beijing with cool temperatures over the mountains. WRF temperatures driven by MIROC-ESM and MIROC-ESM-CHEM are higher in the Beijing and Tianjin city centers than ERA5, ISIMIP-MIROC and ISIMIP-MIROC-CHEM outputs, while the temperatures in the suburbs is a little lower than from ERA5

and ISIMIP-ESM. Temperatures from HadGEM2-ES driven WRF are a little colder than that of ERA5 and ISIMIP-HadGEM2 over Beijing (Fig. S1). WRF also produces lower relative humidity in the center of Beijing and higher humidity in the north and west of city, consistent with the pattern of temperatures. Humidity under WRF tends to be lower in the urban center (45%) and higher in the suburban areas (60%) than ISIMIP and ERA5. Relative humidities of all ESM are higher than ERA5 (Fig. S2). The

humidity of WRF under different ESMs is noticeably different from each other, although MIROC-ESM and MIROC-ESM-CHEM are similar (Fig. S2). Wind speeds in all ESMs are greater than ERA5, except HadGEM2-ES (Fig. S3). ISIMIP reduces all ESM to essentially the ERA5 pattern as with temperature and humidity. WRF winds for all four ESM are greatly overestimated. All WRF simulations have spatial patterns very different from ERA5, with maximums associated with the

northern and western higher ground.





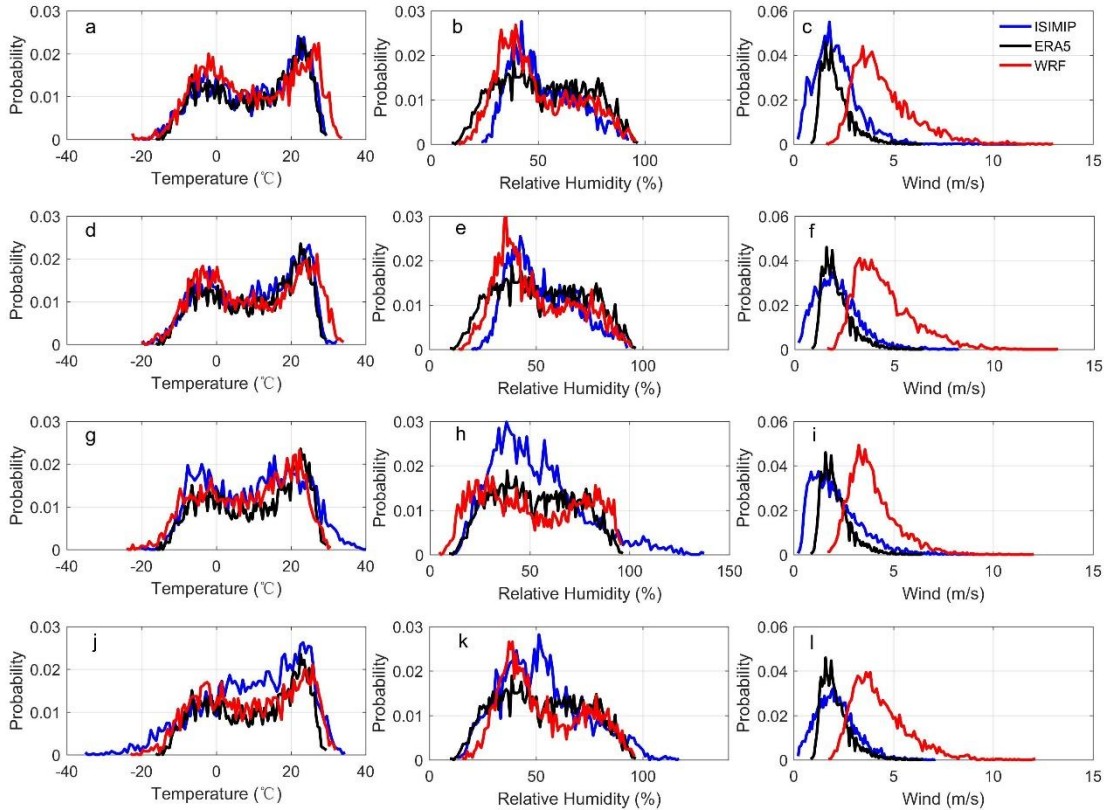

**Figure 4.** The probability density function (pdf) for daily mean temperature, relative humidity and wind speed for MIROC-ESM (a-c), MIROC-ESM-CHEM (d-f), HadGEM2-ES (g-i) and BNU-ESM (j-l) under WRF (red lines) and ISIMIP statistical bias correction method (blue lines) in the Beijing province (Fig. 1) during 1980-1989. The black line is ERA5 reanalysis data.

The temperature, relative humidity and windspeed distributions (Fig. 4) illustrate bias and across-ESM differences for the WRF simulations. Both MIROC-ESM and MIROC-ESM-CHEM overestimate the probability of high temperatures. ISIMIP-HadGEM2 overestimates likelihood of high temperatures compared with ERA5, while ISIMIP-BNU overestimates both high and low temperatures extremes. WRF performs well for all four ESM compared with ISIMIP which produces unphysical relative humidities exceeding 100% for HadGEM2-ES and BNU-ESM. ISIMIP winds for all four ESMs tend to increase the frequency of low winds and winds exceeding 5 m/s. WRF winds are close to twice that of ERA5. Overall, results after ISIMIP shows closer mean values to ERA5 while the pdfs for WRF are closer to ERA5, but the differences between ISIMIP and WRF are small except for wind speed.



**Figure 5.** Taylor diagram for daily temperature (a), wind speed (b) and relative humidity (c) of four ESMs using two downscaling methods, i.e., ISIMIP (red) and WRF (brown) compared to ERA5 data during 1980-1989 in Beijing. The blue symbols are the data from raw ESMs. The skill of downscaling methods is reflected by the distance from each symbol to the point labelled "ERA5", the ERA5 reanalysis data. The blue lines are correlation coefficient which represents the similarity between each downscaling data and reanalysis data. The green contours are root mean standard deviation (RMSD), and black contours are standard deviation.



Fig. 5 shows the Taylor diagrams (Taylor, 2001), which can be used to assess the skill of two downscaling methods applied to meteorological data. Temperatures from raw MIROC-ESM and

MIROC-ESM-CHEM output show better performance than the other two models. WRF has better correlation coefficient (>0.95) than ISIMIP, and smaller RMSD for all four ESMs. Wind speed of all four ESMs outputs have correlation coefficients <0.1 with ERA5. ISIMIP greatly reduces errors and variance but does not improve correlation. WRF has the better correlation, lower errors and shows better skill on simulating the wind speed than ISIMIP, despite its systematic bias in magnitude.

**3.2 Bias correction for WRF**





**Figure 6.** The spatial distribution of mean 2 m temperature (a-c), relative humidity (d-f) and 10 m wind speed (g-i) from ERA5, WRF-ESM and WRF-QDM-ESM multi-model ensemble mean during 2008-2017. Figs. S4-S6 shows the four ESM results separately.

We use QDM to correct biases of the WRF results for the 2008-2017 historical simulation. The temperature, relative humidity and wind speed from ERA5 over Beijing during 2008-2017 (Fig. 6) have a very similar pattern with that during 1980s (Fig. 3). Average temperatures slightly increased over Beijing compared with the 1980s. Average humidities in most places during 2008-2017 are slightly higher than during the 1980s, while in the northwest of Zhangjiakou where temperatures rose fastest,

humidity shows a slight decrease (Fig. 3, Fig. 6). Winds in Beijing between these two decades did not change.

WRF-QDM-ESM simulations of the three variables (Fig. 6) exhibit geographic patterns that are the same as that during the 1980s (Fig. 3). QDM bias correction makes the temperature hotter and the humidity drier especially in high mountains and cities, producing spatial patterns closer to ERA5 (Fig.

6). QDM bias correction greatly improves wind speed results from uncorrected values of 4-5 m s$^{-1}$ to 1.5-2.5 m s$^{-1}$ across most areas. The mean 2 m temperature of WRF-QDM-HadGEM is hotter than the other three ESMs (16°C in the center of Beijing), while the other three reach only 14°C, which is close to ERA5 (Fig. S4e-4h). WRF-QDM-MIROC and WRF-QDM-MIROC-CHEM are more humid than WRF-QDM-HadGEM2 and WRF-QDM-BNU. The wind speed of WRF-QDM-BNU is a little smaller

than other three ESMs.

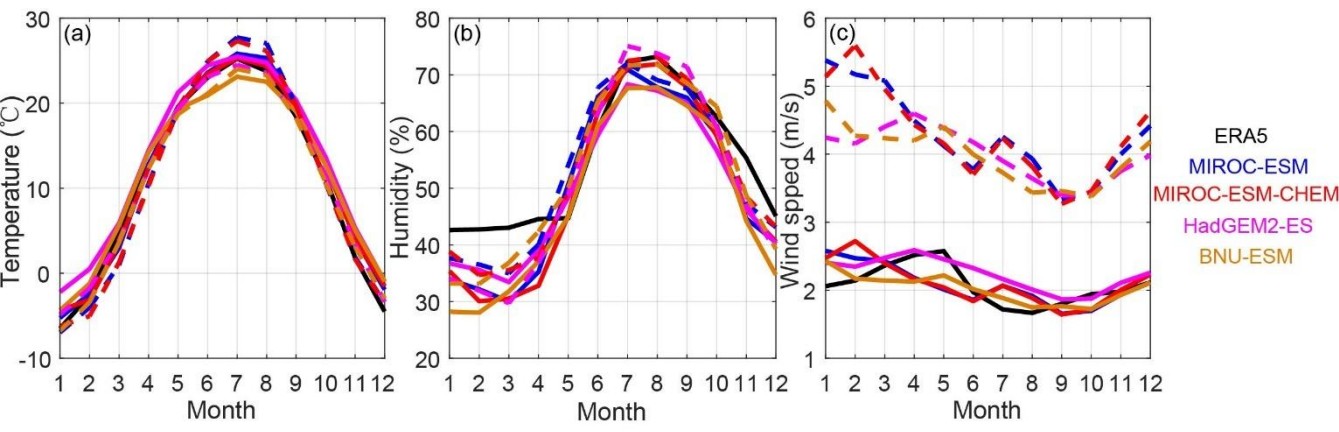



**Figure 7.** Seasonal cycle of multi-year averaged monthly temperature (a), relative humidity (b) and wind speed (c) during 2008-2017 for Beijing. The solid lines are the WRF-QDM-ESM results and the dashed lines are the WRF-ESM results.

The seasonal cycle of average daily temperature simulated by WRF is close to ERA5 (Fig. 7). However, the temperature of WRF-QDM-BNU shows a colder bias than the raw results in the summer and the temperature of WRF-QDM-HadGEM2 shows a warmer bias than the results from WRF-HadGEM2 in the winter. For humidity, the overall performances of the WRF-QDM-ESM are not good and they all show a dry bias relative to ERA5 from Jan-May. During Jun-Oct, the humidity from both WRF-QDM-

MIROC and WRF-QDM-MIROC-CHEM show a wetter pattern than ERA5. After bias correction, the wind speed from all ESMs clearly decreased to the same range as ERA5. Winds from HadGEM2-ES shows the best agreement in both the quantity and seasonality with ERA5 but QDM does not change the seasonality of wind speed, just its amplitude.





**Figure 8.** Taylor diagram for daily temperature (a), relative humidity (b) and wind speed (c) of WRF driven by four ESMs results with QDM bias correction (red) and without bias correction (blue) compared to ERA5 reanalysis data (purple star) during 2008-2017 in Beijing.



The skill of QDM for correcting daily temperature, relative humidity and wind from the biased WRF-ESM are shown in figure 8. QDM changes the errors but has little effect on the correlation with the

ERA5 data. While temperature and wind speed all improve with QDM, humidities from WRF-QDM-ESM has a lower performance than does the corresponding WRF-ESM results except for WRF-QDM-HadGEM2. We regard bias correcting as necessary for WRF outputs.

## 3.2 Future Projections

We now look at temperature, humidity and wind projections for 2060-2069. Fig. 9 shows maps of

ensemble mean 2 m temperature (Fig. 9a-f), relative humidity (Fig. 9g-l) and 10 m wind speed (Fig. 9m-r) under the G4, RCP4.5 and RCP8.5 scenarios. Mean temperatures over the domain are 12.5 ℃, 13.3 ℃ and 14.8 ℃ from ISIMIP-ESM, and 13.1 ℃, 13.8 ℃ and 15.2 ℃ from WRF-QDM-ESM under G4, RCP4.5 and RCP8.5 scenarios respectively. The higher city center temperatures from WRF-QDM-ESM account for the 0.5℃ difference from ISIMIP-ESM. Both ISIMIP-ESM and WRF-QDM-ESM

produce similar overall temperature patterns driven by topography. Relative humidities under geoengineering and RCP scenarios are almost the same. Mean model relative humidities are 53.8%, 53.6% and 53.4% by ISIMIP-ESM, and slightly wetter than 50.0%, 49.7% and 50.2% from WRF-QDM-ESM, under G4, RCP4.5 and RCP8.5 scenarios respectively. This is mainly due to lower humidities with WRF-QDM-ESM in city centers. Wind spatial patterns are clearly different from

ISIMIP-ESM and WRF-QDM-ESM. The windspeed in the southwest and southeast of the domain from ISIMIP-ESM is low, while WRF-QDM-ESM winds are lowest in the city center. Although there are some differences in details between different ESM, the overall results are similar as those of ensemble means (Figs. S7-S12).





**Figure 9.** The spatial distribution of ensemble mean 2 m temperature (a-f), relative humidity (g-l) and 10 m wind speed (m-r) under G4, RCP4.5 and RCP8.5 scenarios based on ISIMIP-ESM and WRF-QDM-ESM results during 2060-2069.

Fig. 10 shows temperature, humidity and wind anomalies from WRF-QDM-ESM and ISIMIP-ESM simulations. The mean temperature in the 2060s under G4 is 1-2 ℃ higher than that during 2008-2017. Temperatures from both ISIMIP-ESM and WRF-QDM-ESM are cooler over the whole domain under G4 than that under RCP4.5 by 0.1~1.5 ℃, while there is a larger cooling effect of 1.6~2.8 ℃ under G4 relative to RCP8.5 (Table 3). There is large across-model spread with the two MIROC models having smaller differences (G4-RCP4.5) than the other two models (Table 3), and the two MIROC models show larger differences from each other with ISIMIP-ESM than with WRF-QDM-ESM. Relative humidity anomalies exhibit large differences under different scenarios for ISIMIP-ESM and WRF-QDM-ESM. G4 humidity from ISIMIP-ESM shows a slight reduction of 1 percentage point relative to the 2010s over Beijing, while there is an increment of similar magnitude from the WRF-QDM-ESM results. When compared to RCP4.5 scenario, the humidity under G4 from ISIMIP-ESM shows a slight (1 percentage point) increase, but that from WRF-QDM-ESM shows no statistically significant change. The differences of relative humidities from WRF-QDM-ESM and ISIMIP-ESM between G4 and RCP8.5 show opposite trends although differences are slight. ISIMIP-ESM winds under G4 are a little smaller than that during 2010s and show no significant difference between G4 and RCP4.5. Compared to RCP8.5, G4 winds from ISIMIP-ESM increase by 0.15 m s$^{-1}$ mainly in the south of the domain. Winds from WRF-QDM-ESM show very small changes with slight increases relative to the RCP scenarios. Humidity and windspeed anomalies from ISIMIP appear somewhat spatially anti-correlated, while for WRF there are no particular patterns.



**Table 3.** Difference of 2 m temperature between G4 and ERA5 in the 2010s and RCP scenarios in the 2060s for the high resolution domain (Fig. 1). Bold indicates the differences or changes are significant at the 95% confidence level according to the Wilcoxon signed rank test. (Units: °C)

| | G4-2010s | | G4-RCP4.5 | | G4-RCP8.5 | |
| --- | --- | --- | --- | --- | --- | --- |
| | ISIMIP | WRF | ISIMIP | WRF | ISIMIP | WRF |
| MIROC-ESM | **1.9** | **2.0** | **-0.9** | -0.4 | **-2.3** | **-1.9** |
| MIROC-ESM-CHEM | **2.4** | **2.0** | -0.1 | -0.4 | **-1.6** | **-1.7** |
| HadGEM2-ES | **1.0** | **0.8** | **-1.4** | **-1.5** | **-2.8** | **-2.7** |
| BNU-ESM | **1.1** | **1.0** | **-1.2** | **-0.7** | **-2.6** | **-2.2** |
| Ensemble | **1.6** | **1.5** | **-0.9** | **-0.7** | **-2.3** | **-2.2** |









**Figure 10.** Spatial pattern of ensemble mean 2 m temperature (a-f), relative humidity (g-l) and 10 m
wind (m-r) scenario differences: G4-2010s (left column), G4-RCP4.5 (middle column) and G4-RCP8.5
(right column) based on ISIMIP-ESM and WRF-QDM-ESM results. 2010s means the results simulated
during 2008-2017, and G4, RCP4.5 and RCP8.5 means the results projected during 2060-2069.
Stippling indicates grid points where differences are not significant at the 95% confidence level
according to the Wilcoxon signed rank test. Figs. S13-18 show the results for each ISIMIP-ESM and
WRF-QDM-ESM separately.

## 4 Discussion and Conclusions

We have explored the impact of geoengineering on surface temperature, humidity and windspeed over
Beijing during 2060-2069 using statistical bias-correction and dynamical downscaling. We evaluated
the performance of ISIMIP and WRF methods during 1980-1989 based on the historical simulations
from four ESMs. WRF output needs to be bias-corrected for it to be comparable with observations or
with statistically downscaled and bias-corrected output. We use the QDM method to correct the bias in
WRF results since QDM ensures that the pdf of simulation results is consistent with the reanalysis data.
Because we want to keep the high spatial resolution of the WRF model simulation, we do not correct
biases grid cell by grid cell, which would produce output at the reanalysis resolution, but instead doing
bias correction for daily mean temperature, humidity and wind speed over the domain.

The raw output from the ESM of temperature, humidity and wind speed, have no clear spatial
distribution over the domain because cities occupy only a few ESM grid cells. Statistical bias correction
and downscaling by the ISIMIP method produces output at the same resolution as the observational
reanalysis data and matches its spatial distribution. The ISIMIP method is designed to preserve trends
and the long-term mean value the same as observations (Hempel et al., 2013). Dynamically
downscaling demands a higher resolution grid, and WRF produces an output at a spatial resolution
independent of the resolution of the reanalysis data. WRF results not only produce characteristics
consistent with the reanalysis data, but also depict the more detailed meteorological characteristics
created by the complex underlying land surface that are input to WRF. The WRF simulation during
1980-1989 showed higher temperatures in the city centers than that in the suburbs and lower
temperatures in western and northern mountainous areas. This pattern is created by the joint action of





latitude, terrain and underlying surface. The pattern of relative humidity distribution is anticorrelated to temperature, with lower humidity in the urban center, while humidities in some mountainous areas are

higher. This is similar to the pattern of humidity from 42 AWS in Beijing during 2007-2015 (Yang et al., 2017), and due mainly to the underlying surface, and the transpiration of the urban area being less than that of the suburbs (Dou et al., 2020). Windspeeds inside the Beijing urban area are low but reach a maximum in its western foothills. Urbanization increases surface roughness, lowering windspeed (Liu et al., 2020). The pdf of temperature suggests that the WRF result is more realistic and closer to

observations than results from ISIMIP, although the MIROC-ESM and MIROC-ESM-CHEM show a higher probability for high temperatures. The pdf of humidity strongly indicates that WRF performs better than ISIMIP. But WRF tends to overestimate the wind speed even though the shape of pdf is more similar to observations than that of ISIMIP output. Many studies show that WRF frequently overestimate the wind speed, e.g., in the Gulf of Mexico (Lee et al., 2011), in the coastal cities of Spain

(Chen et al., 2012), and in south-eastern Texas (Ngan et al., 2013) because of imperfect surface representation (e.g., urban vegetation and surface morphology). Overall, for temperature and humidity, the results of WRF are better than those of ISIMIP, and both are better than the original ESM output. WRF has better correlation with observations than ISIMIP.

Applying QDM bias correction to WRF reduces model and monthly dependent differences. Temperatures, both before and after bias correction, have high correlations with ERA5 and QDM makes little difference. WRF relative humidity, however, is always wetter than that of ERA5 in winter whether revised or not. Wind speeds are lowered after bias correction to the same levels as ERA5, but QDM does not have a clear effect on correlation (Zhao et al., 2017).

The spatial distribution of temperature, humidity and wind speed are roughly similar in all three periods assessed, that is the 1980s, 2010s and 2060s, whether from ISIMIP-ESM or WRF-QDM-ESM. Our analysis shows that mean temperature under G4 SAI is always lower than that under RCP4.5 and RCP8.5. Although it does not return temperatures to the historical level, that was not the design of the experiments which instead simply explore the effects of injecting roughly ¼ the amount of $SO_2$ into the

equatorial lower stratosphere as the 1991 Mt. Pinatubo eruption every year for 50 years. Using ISIMIP
       downscaling leads to larger differences between scenarios than using WRF. HadGEM2-ES shows the
       largest difference in temperatures between G4 and RCP scenarios of the four ESM we study. For the
       relative humidity, ISIMIP-ESM and WRF-QDM-ESM give opposite (but small) signed anomalies
       between G4 and RCP4.5 in our domain. The 2010s were slightly less humid according to WRF-QDM-
ESM than G4 in the 2060s while they were a little wetter according to ISIMIP-ESM.

       This paper is the first to use WRF for regional dynamic downscaling of geoengineered climates and
       impacts on relatively small spatial and temporal scales that can be useful for regions that need higher
       resolutions than ERA5 and statistical downscaling can supply. The differences between statistical
       downscaling and dynamic downscaling in the Beijing provincial region, that extends from sea level to
mountains about 2000 m in elevation, may appear rather modest. But even these modest differences in
       derived temperatures and humidities can make for large differences in compound indices such as
       apparent temperature, particularly when assessing future risk to urban populations.

**Code and data availability**

       All ESM data used in this work are available from the Earth System Grid Federation (WCRP, 2021;
https://esgf-node.llnl.gov/projects/cmip6, last access: 14 July 2021). The WRF and ISIMIP bias-
       corrected and downscaled results are available for the authors on request. WRF and ISIMIP codes are
       freely available at the references cited in the methods sections.

**Supplement link**

       the link to the supplement will be included by Copernicus.

**Author contribution**

       JCM and LZ designed the experiments, JW performed the simulations. JW and JCM prepared the
       manuscript with contributions from all co-authors.





## Competing interests

The authors declare that they have no conflict of interest.

## Disclaimer

Publisher's note: Copernicus Publications remains neutral with regard to jurisdictional claims in published maps and institutional affiliations.

## Special issue statement:

This article is part of the special issue "Resolving uncertainties in solar geoengineering through multi-
model and large-ensemble simulations (ACP/ESD inter-journal SI)". It is not associated with a conference.

## Acknowledgements

We thank the climate modeling groups for participating in the Geoengineering Model Intercomparison Project and their model development teams; the CLIVAR/WCRP Working Group on Coupled
Modeling for endorsing the GeoMIP; and the scientists managing the earth system grid data nodes who have assisted with making GeoMIP output available. Ben Kravitz provided useful advice on the manuscript. This research was funded by the National Key Science Program for Global Change Research (2015CB953602)

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
