# Peer review of "Regional dynamical and statistical downscaling temperature, humidity and windspeed for the Beijing region under stratospheric aerosol injection geoengineering"

_Earth System Dynamics, 2022_

## Author Comment (AC1)

Referee's comments are in red, our reply in black, quotes in the revised manuscript in blue.

**Referee 1's comments**

General comments

Wang et al. presents an interesting study about comparison between two downscaling approaches i.e. statistical downscaling (ISIMIP) and dynamical downscaling (WRF) combined with bias correction of the future projection of ESM scenarios (i.e. RCP4.5, RCP8.5, G4) over Beijing provincial region. The idea of comparing statistical with dynamical downscaling is a novel path of research. The authors focus on mean temperature, humidity, and wind speed, which are all relevant for climate impacts. The manuscript is generally clearly written and well structured, and the analysis is done in an organized way. However, some parts need to be clarified to easily understand some contents of the manuscript. I think the manuscript is scientifically sound and merits publication after some minor revisions based on my specific comments below.

Reply: We thank the reviewer for their overall positive and very constructive response, along with their helpful suggestions for improving the manuscript. Our response is given below.

Specific comments:

1. In table 1, assessment on the performance of ISIMIP and WRF with bias correction is done only through RCP4.5 scenario. Is there any reason why RCP8.5 is excluded in this part?

Reply: In the assessment part, we only want to see the performance of QDM methods. So we choose the RCP4.5 scenario during 2008-2017 as our reference to simulate. There is no statistical difference between RCP4.5 and RCP8.5 in the 2007-2017 period so it does not matter whether we use RCP4.5 or RCP8.5.

2. This research used ERA5 reanalysis data as the proxy of observation. The authors need strong justification of choosing ERA5 reanalysis in this research as reanalysis data itself might contain some degree of bias.

Reply: Thanks for your comment. Choosing a reanalysis data is an important thing. ERA5 has a finer temporal resolution (hourly) and spatial resolution (0.25°). We add these sentences in Section 2.1

[revised manuscript text omitted]

4.The statement "QDM is similar to QM but is non-stationary". In what sense it is non-stationary?

Reply: Non-stationary means that the mean value or the probability density function (pdf) of a time series changes over time. QM is used to correct the pdf of model data under any scenario to be consistent with the pdf of observed data in the reference period. But the pdf in the future scenario is not the same as in the historical reference period. Therefore, QDM takes the difference between the pdf of model data and observed data in the reference period into account, and applies this difference to the

model data in the future scenario. In this sentence, what we want to express is that QM assumes that the pdf of model data does not change over time (under different scenarios).

5.From figure 3i we see that the WRF simulations have spatial patterns very different from Moreover, figure 4 also shows that the pdf of WRF deviates from ERA5 significantly. Are there any reasons behind these findings?

Reply: From the figure 3i and 4, we can see that WRF overestimates the wind speed. In our paper discussion part (line 402-403 and 408-411), we only mentioned the reason for distribution of wind speed from WRF results. A new surface drag parameterization scheme was launched after WRFv3.4, although we chose not to use it here. Zheng et al. (2016) evaluated the effect of two surface drag parameterization schemes on the surface wind speed in Beijing, and found that terrain correction can improve the simulation of wind speed in valley areas. However, WRF still has a systematic overestimation of wind speed. We add the following sentence in the discussion section.

Overestimated surface wind speed in WRF is caused by using smoothed topography in the model (Jimenez and Dudhia, 2012).

Reply: Sorry we prefer to use bold – as is the case on many articles. But perhaps it can be left to the journal style and editor decision on what approach to take.

---

## Author Comment (AC2)

Referee's comments are in red, our reply in black, quotes in the revised manuscript in blue.

**Referee 2's comments**

General Comments:

This paper looks at statistical and dynamical downscaling for the Beijing region under three future climate scenarios. It also appears to be the first time that dynamical downscaling has been used with a geoengineering scenario. The paper is fairly straightforward but seems to be a good starting point for future work on geoengineering modeling. Probably the biggest question I have is regarding the choice of analyzed variables. The authors compare temperature, humidity, and wind speeds under the different scenarios; temperature is of course important, but, at least to me, the latter two are not so much. I wonder if the authors thought about analyzing precipitation instead.

Reply: As you mentioned, the effects of SAI on precipitation is also very important. Precipitation has often been studied and in general changes have been widely reported, not least in our own work e.g. in a ¼° global food and streamflow analysis (Wei et al 2018 in *ACP* doi: 10.5194/acp-18-16033-2018). But in this analysis, we wanted to focus on the meteorological elements that impact human health. Temperature, humidity and wind speed are factors that affect apparent temperature, especially in extreme events for big cities. Humidity and windspeed have been far less studied in geoengineering scenarios than precipitation, and so does merit some analysis.

Another general question I have is: how is the ISIMIP spatial downscaling done? The two steps in Section 2.3 address bias correction and temporal downscaling, but how do you go from the coarse ESM grids to the ERA5 grid?

Reply: We have rewritten the Section 2.3 in more detail.

Step 1: We firstly bilinearly interpolate the model data to the same grid points of reanalysis data before bias correction.

Step 2: Monthly bias corrected data are found by multi-year averaged difference between the model output and reanalysis data in our referenced period.
$$M_m^*=\overline{R_m}-\overline{M_m}+M_m \qquad\qquad (2)$$
The $M_m^*$ is the bias-corrected monthly data, $\overline{R_m}$ and $\overline{M_m}$ are the multi-year averaged values in this month from reanalysis data and model data during the reference period, respectively. $M_m$ is the modeled monthly data. The subscript $m$ represents monthly. In this step, ISIMIP does not correct the daily variability of modeled data.
Step 3: Correct the modeled daily variability to a linear regression residual.
$$\Delta M_d^*=\overline{B}\times(M_d-M_m) \qquad\qquad (3)$$

The $\Delta M_d^*$ is the bias-corrected residual daily data from model. $M_d$ is the modeled daily data The subscript $d$ represents daily. $(M_d\text{-}M_m)$ represents the modeled daily residual values in this month, and residual of reanalysis data can be obtained in the same way. $\bar{B}$ is the linear regression coefficient of daily residual values between reanalysis data and model data during our referenced period. Then, we can get the bias-corrected modeled daily data:

$$M_d^* = M_m^* + \Delta M_d^* \qquad\qquad\qquad (4)$$

The $M_d^*$ is the bias-corrected daily data of model. Therefore ISIMIP corrects the monthly mean and its daily variability. Here, we use the ERA5 reanalysis data as reanalysis data in our study. For convenience we use the term ISIMIP-ESM to denote the output from the ESMs after applying the ISIMIP statistical downscaling and bias correction methodology.

Finally, why is quantile delta mapping not done for the historical WRF simulations (in which case I note that it would simplify to the regular quantile mapping method)? Wouldn't it be fairer to compare ERA5 with bias-corrected ISIMIP/statistical downscaling and bias-corrected WRF/dynamical downscaling?

Reply: In our historical simulation, we only want to see the performance of ESM raw data, ISIMIP results and WRF results – without bias correction. The performance of WRF on the temperature and humidity is good in our historical simulation, but for the wind speed WRF shows an overestimation. If WRF showed good performance on all three variables, we would not need to do bias correction for WRF results, but the analysis shows that we do in fact need to bias correct the WRF output. We think this is a worthwhile point to make, and we do compare the bias corrected WRF with ERA5 and uncorrected WRF in fig. 6.

Specific Comments:

Abstract, line 20: It is not clear to me what "larger spatial ranges" means. I also do not see further reference to this in the main text. While I can see from Figure 9 that the range (maximum minus minimum) of temperature and of humidity is larger in WRF than ISIMIP, it's not obvious that this is the case for wind since although WRF has lower minima, ISIMIP has higher maxima.

Reply: This sentence refers to temperature ranges across the Beijing province. We have rewritten this sentence.

In the 2060s WRF produces consistently larger spatial ranges of surface temperatures, humidities and windspeeds than ISIMIP downscaling across the Beijing province for all three future scenarios.

pg 3, line 60: Give the full name of ISIMIP at least once (perhaps in the abstract as well).

Reply: Done. We have rewritten this sentence.

The Inter-Sectoral Impact Model Intercomparison Project (ISIMIP, https://www.isimip.org/) consortium has produced methods (Hempel et al., 2013) widely used to correct the bias from CMIP5 (Climate Model Intercomparison Project phase 5) and GeoMIP outputs (McSweeney et al., 2016; Moore et al., 2019; Kuswanto et al., 2021).

pg 12 / Figure 4: Are the authors concerned about RH values exceeding 100%?

Reply: In Fig. 4, RH from ISIMIP-HadGEM and ISIMIP-BNU do have some values exceeding 100%. Hempel et al. (2013) which defines the ISIMIP method, avoids negative values for radiation and wind speeds but it does not mention that the humidity may exceed 100%. So, the method has a systematic problem. But we note that the fraction of data concerned is small. We have add this to the fig. 4 caption:

Values of humidity exceeding 100% can occur with ISIMIP downscaling.

pg 14, line 277: ISIMIP seems to increase error and variance in wind speed in the case of HadGEM2-ES.

Reply: Yes, we have rewritten this sentence

ISIMIP greatly reduces errors and variance (except for HadGEM2-ES) but does not improve correlation.

pg 16, line 309-310: Assuming I'm looking correctly, it doesn't look like WRF-QDM-MIROC and WRF-QDM-MIROC-CHEM are wetter than ERA5 from June to October. If anything, they look drier, especially the former.

Reply: Yes, we have deleted that statement and rewritten this sentence.

For humidity, the overall performances of the WRF-QDM-ESM are not good and they all show a dry bias relative to ERA5 from July to the following May.

pg 18, line 335-336: Wind speeds in ISIMIP look highest (as opposed to lowest) in the southeast of the domain.

Reply: Yes. We have deleted the southeast.

The windspeed in the southwest of the domain from ISIMIP-ESM is low, while WRF-QDM-ESM winds are lowest in the city center.

pg 20, line 360: To me, only the G4-RCP8.5 anomalies for ISIMIP look anti-correlated. I might suggest removing this sentence.

Reply: We have rewritten this sentence.

Humidity anomalies from ISIMIP have a difference under G4 relative to RCP8.5 in the southwest of the domain, where windspeed anomalies show an obvious positive change, while for WRF there are no particular patterns.

pg 24, line 417: I think the authors meant to say that WRF relative humidity exhibits a dry bias in winter.

Reply: Yes, we have rewritten this sentence.

WRF relative humidity, however, is always drier than that of ERA5 in winter whether revised or not.

pg 25, line 429-430: This sentence is confusing as written, and the first part is not obvious since WRF-QDM-ESM shows opposite trends in roughly equal parts of the domain when comparing G4 and the 2010s. I might suggest removing this sentence.

Reply: Done.